# GEOMSTATS: A PYTHON PACKAGE FOR RIEMANNIAN GEOMETRY IN MACHINE LEARNING

## ABSTRACT

We introduce `geomstats`, a Python package for Riemannian modelization and optimization over manifolds such as hyperspheres, hyperbolic spaces, spaces of SPD matrices or Lie groups of transformations. Our contribution is threefold. First, `geomstats` allows the flexible modeling of many a machine learning problem through an efficient and extensively unit-tested implementations of these manifolds, as well as the set of useful Riemannian metrics, exponential and logarithm maps that we provide. Moreover, the wide choice of loss functions and our implementation of the corresponding gradients allow fast and easy optimization over manifolds. Finally, `geomstats` is the only package to provide a unified framework for Riemannian geometry, as the operations implemented in `geomstats` are available with different computing backends (`numpy`, `tensorflow` and `keras`), as well as with a GPU-enabled mode– thus considerably facilitating the application of Riemannian geometry in machine learning. In this paper, we present `geomstats` through a review of the utility and advantages of manifolds in machine learning, using the concrete examples that they span to show the efficiency and practicality of their implementation using our package[1].

## 1 INTRODUCTION

From soft-classification to image recognition, Riemannian manifolds provide a natural framework to many a machine learning problem. Consider the following standard supervised learning problem: given an input $X$, the goal is to predict an output $Y$. The relation between $X$ and $Y$ is typically modeled by a function $f_\theta : X \to Y$, characterized by a set of parameters $\theta$. Riemannian geometry often naturally arises at each of the three different stages of the modelization: through the input $X$, the output $Y$, or the parameters $\theta$. For instance, the input $X$ might belong to a Riemannian manifold. This is typically the case in image processing, where images $X$ are frequently modeled as elements of a low-dimensional manifold (17; 57; 61; 67; 18; 6). Such is the case in (62), in which the authors consider spherical images as elements of the orthogonal rotation group SO(3). In some cases, $X$ can even be a manifold itself— in (8) for instance, the authors propose to model images as a function of a 2D smooth surface representing a shape such as a human pose. Similarly, the output $Y$ often belongs to a Riemannian manifold (42; 47). Such is the case in problems where the output is a member of the set of doubly stochastic matrices –as for instance in some neurosciences applications (48; 11)—. or when the optimization is carried on a given manifold (2; 44; 27). In (28) for example, the authors use a neural network to predict the pose of a camera $Y$, which is defined as an element of the Lie group $SE(3)$. Finally, the parameter $\theta$ of a model can be constrained on a Riemannian manifold, such as in the work of (29) which constrains the weights of a neural network on multiple dependent Stiefel manifolds.

Manifolds offer intuitive and practical advantages for modeling inputs, outputs and parameters. When applied to the input, they constitute lower dimensional spaces with fewer degrees of freedom, thus potentially allowing faster computations and less substantial memory allocation costs. Moreover, the non-linear degrees of freedom in these manifolds are often more intuitive and benefit from more expressive power. For instance, the geolocation of points of interest on Earth is more efficiently achieved through their longitude and latitude—i.e., their 2D manifold coordinates—rather than through their position $(x, y, z)$ in the 3D Cartesian space. Most current machine learning problems make little use of this underlying manifold structure —rather viewing their optimization task as a

---

[1]The package is provided in a zip file at the following anonymized address: `https://goo.gl/XV2Rb7`.

constrained optimization over Euclidean space. Riemannian geometry, on the other hand, attempts to leverage the manifold structure to solve the corresponding optimization problem, replacing lines by geodesics, partial differential by covariate differentiation (59)— thus potentially reducing the dimension of the prblem space and the memory allocation costs.

Yet, the adoption of Riemannian geometry by the machine learning community has been largely hindered by the lack of a modular framework for implementing such methods. Code sequences are often custom tailored for specific problems and/or computing backends, and are thus not easily re-usable: . To address this issue, some packages have been written to perform computations on manifolds. The `theanogeometry` package (39) provides an implementation of differential geometric tensors on manifolds where closed forms do not necessarily exist, using the automatic differentiation tool `theano` to integrate differential equations that define the geometric tensors. The `pygeometry` package (10) offers an implementation primarily focused on the Lie groups $SO(3)$ and $SE(3)$ for robotics applications. However, no implementation of non-canonical metrics on these Lie groups is provided. The `pymanopt` package (63), originally implemented in Matlab as `manopt`, provides a very comprehensive toolbox for optimization on a extensive list of manifolds. However, not only is the choice of metrics on these manifolds rather restricted, the manifolds themselves are often implemented using canonical embeddings in higher-dimensional euclidean spaces, with high computational costs.

This paper presents `geomstats`, a package specifically targeted at the machine learning community to perform computations on Riemannian manifolds with a flexible choice of Riemannian metrics. The `geomstats` package makes three contributions. First, `geomstats` is the first Riemannian geometry package to be extensively unit-tested with more than 90 % code coverage. Second, `geomstats` implements `numpy` (51) and `tensorflow` (1) backends, providing vectorized, intuitive computations, and available for GPU implementation. We also provide an updated version of the deep learning framework,`keras`, equipped with Riemannian gradient descent on manifolds. Finally, `geomstats` strives to be a user-friendly and educational a tool, presenting Riemannian geometry to computer scientists and facilitating its use as a *complement* to theoretical papers or books. We refer to (55) for the theory and expect the reader to have a high-level understanding of Riemannian geometry.

Our paper is organized as follows. We begin by providing an overview of `geomstats` in Section 2. We then present concrete use cases of `geomstats` for machine learning on manifolds of increasing geometric complexity, starting with manifolds embedded in flat spaces in Section 3, to a manifold embedded in a Lie group with a Lie group action in Section 4, to the Lie groups $SO(n)$ and $SE(n)$ in Section 5. Along the way, we present a review of the occurrences of each manifold in the machine learning literature, some educational visualizations of the Riemannian geometry as well as implementations of machine learning models where the inputs, the outputs and the parameters successively belong to manifolds.

## 2 THE PACKAGE GEOMSTATS

### 2.1 GEOMETRY

The `geomstats` package implements Riemannian geometry through a natural object-oriented approach based on two main families of classes: the manifolds, inherited from the class `Manifold` and the Riemannian metrics, inherited from the class `RiemannianMetric`. Children classes of `Manifold` considered here include: `LieGroup`, `EmbeddedManifold`, `SpecialOrthogonalGroup`, `SpecialEuclideanGroup`, `Hypersphere`, `HyperbolicSpace` and `SPDMatricesSpace`. Once the user has specified an instance of `Manifold`, she or he must endow it with a particular choice of Riemanniann metric; Instantiations of the `RiemannianMetric` class and its children classes are attributes of the manifold objects.

The class `RiemannianMetric` implements the usual methods of Riemannian geometry, such as the inner product of two tangent vectors at a base point, the (squared) norm of a tangent vector at a base point, the (squared) distance between two points, the Riemannian Exponential and Logarithm maps at a base point and a geodesic characterized by an initial tangent vector at an initial point or by an initial point and an end point. Children classes of `RiemannianMetric` include the class `InvariantMetric`, which implements the left—and right—invariant metrics on Lie groups,as well as `EuclideanMetric` and `MinkowskiMetric` which are respectively the most standard flat Riemannian and pseudo-Riemannian metrics.

Contrary to prior existing packages, our methods have been extensively unit-tested, with more than 90% code coverage. The code is provided with `numpy` and `tensorflow` backends. Moreover, `geomstats` strives for efficiency: the code is vectorized through the use of arrays in order to facilitate intuitive batch computation. The `tensorflow` backend also enables running the computations on GPUs.

## 2.2 STATISTICS AND MACHINE LEARNING

The package `geomstats` also allows the user to easily perform statistical analysis on manifolds —specifically for Riemannian statistics— in the class `RiemannianMetric` (53). The class `RiemannianMetric` implements the weighted Fréchet mean of a dataset through a Gauss-Newton gradient descent iteration (21), the variance of a dataset with respect to a point on the manifold, as well as tangent principal component analysis (20). This provides an easily and readily-applicable tool for analysts to investigate and define summary statistics from the Riemannian perspective.

From a machine learning viewpoint, `geomstats` strives to facilitate the use of Riemannian geometry in machine learning and deep learning settings by providing easy and practical frameworks to incorporate manifold constraints into the optimization objective. Let us suppose for instance that we want to train a neural network to predict an output $Y$ on the manifold of our choice. `Geomstats` provides off-the-shelf loss functions on Riemannian manifolds, implemented as squared geodesic distances between the predicted output and the ground truth. These loss functions are consistent with the geometric structure of the Riemannian manifold. The package gives the closed forms of the Riemannian gradients corresponding to these losses, so that back-propagation can be easily performed. Let us suppose now that we want to constrain the parameters of a model, such as the weights of a neural network, to belong to a manifold. We provide modified versions of `keras` and `tensorflow`, so that they can constrain weights on manifolds during training. As such, the availability of `geomstats` from many different computing backends, as well as its computational efficiency should greatly facilitate the use of Riemannian geometry in machine learning/

In the following sections, we demonstrate the use of the manifolds implemented in `geomstats`. For each manifold, we present a literature review of its appearance in machine learning and we describe its implementation in `geomstats` along with a concrete use case.

## 3 EMBEDDED MANIFOLDS - HYPERSPHERE AND HYPERBOLIC SPACE

We begin by considering the hypersphere and the hyperbolic space, often considered as the simplest and most standard Riemannian manifolds of constant curvature (55). The manifolds are respectively implemented in the classes `Hypersphere` and `HyperbolicSpace`. The logic of the Riemannian structure of these two manifolds is very similar. They are both manifolds defined by their embedding in a flat Riemannian or pseudo-Riemannian manifold.

To be more specific, the $n$-dimensional hypersphere $S^n$ is defined by its embedding in the $(n + 1)$-Euclidean space, which is a flat Riemannian manifold, as

$$S^n = \left\{ x \in \mathbb{R}^{n+1} : x_1^2 + ... + x_{n+1}^2 = 1 \right\}. \tag{1}$$

Similarly, the $n$-dimensional hyperbolic space $H_n$ is defined by its embedding in the $(n + 1)$-dimensional Minkowski space, which is a flat pseudo-Riemannian manifold, as

$$H_n = \left\{ x \in \mathbb{R}^{n+1} : -x_1^2 + ... + x_{n+1}^2 = -1 \right\}. \tag{2}$$

The classes `Hypersphere` and `HyperbolicSpace` therefore inherit from the class `EmbeddedManifold`. They implement methods such as: conversion functions from intrinsic $n$-dimensional coordinates to extrinsic $(n + 1)$-dimensional coordinates in the embedding space (and vice-versa); projection of a point in the embedding space to the embedded manifold; projection of a vector in the embedding space to a tangent space at the embedded manifold.

The Riemannian metric defined on $S^n$ is derived from the Euclidean metric in the embedding space, while the Riemannian metric defined on $H^n$ is derived from the Minkowski metric in the embedding space. They are respectively implemented in the classes `HypersphereMetric` and `HyperbolicMetric`.

### 3.1 HYPERSPHERE - REVIEW OF USE CASES IN MACHINE LEARNING

The hypersphere naturally appears in a number of settings. In particular, hyperspheres are common objects in circular statistics (31), directional statistics (45) or orientation statistics (16), that is, areas which focus on the analysis of data on circles, spheres and rotation groups. Concrete applications are extremely diverse and range from biology to physics (49; 37; 32) or trajectory analysis (9; 40) among many others (46; 65). In biology for instance, the sphere $S^2$ is used in nalysis of protein structures (36). In physics, the semi-hypersphere $S_+^4$ is used to encode the projective space $P_4$ for representing crystal orientations in applied crystallography (56; 14).

The shape statistics literature (34) is also manipulating data on abstract hyperspheres. For instance, in (35), the author studies shapes of $k$ landmarks in $m$ dimensions and introduces the "pre-shape" spaces which are hyperspheres $S^{m(k-1)}$. The s-rep, a skeletal representation of 3D shapes, also deals with hyperspheres $S^{3n-4}$ as the object under study is represented by $n$ points along its boundary (26).

Lastly, hyperspheres can be used to constrain the parameters of a machine learning model. For example, training a neural net with parameters constrained on a hypersphere has been shown to result in an easier optimization, faster convergence and comparable (even better) classification accuracy (43).

### 3.2 GEOMSTATS USE CASE - OPTIMIZATION AND DEEP LEARNING ON HYPERSPHERES

We now demonstrate how to use `geomstats` for constraining a neural network's weights on manifolds during training, as advocated in (43). A detailed and reproducible implementation of this example can be found in the `deep_learning` folder of our package.

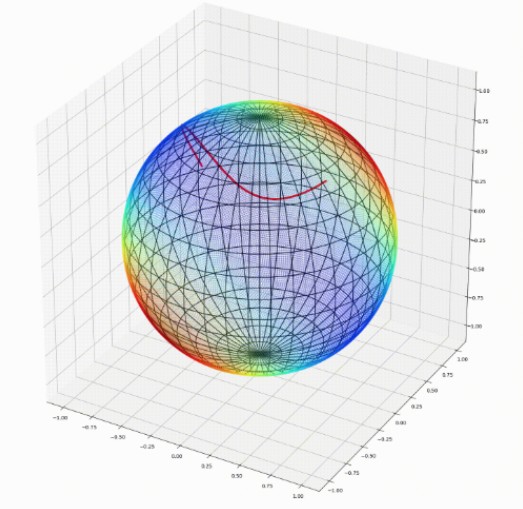

Figure 1: Minimization of a scalar field on the sphere $S^2$. The color map indicates the scalar field values, where blue is the minimum and red the maximum. The red curve shows the trajectory taken by the Riemannian gradient descent, which converges to a minimum (blue region).

This setting requires to first implement Riemannian gradient descent on the hypersphere, a process which we demonstrate in figure 1 on a toy example: here, the goal is to minimize a quadratic form $x^T A x$ with $A \in \mathbb{R}^{n \times n}$ and $x^T A x > 0$ constrained on the hypersphere $S^{n-1}$. `Geomstats` allows us to conveniently generate a positive semidefinite matrix by doing a random uniform sampling on the `SPDMatricesSpace` manifold. The red curve in figure 1 shows the trajectory of the algorithm along the hypersphere.

Coming back to our example, in order to train the neural network's weight on the hypersphere, we have modified the optimization step in `keras`, so that the stochastic gradient descent is done on the manifold through the Exponential map. In our implementation, the user can pass a `Manifold` parameter to each neural network layer. The stochastic gradient descent optimizer has been modified

to operate the Riemannian gradient descent in parallel. It infers the number of manifolds directly from the dimensionality by finding out how many manifolds are needed in order to optimize the number of kernel weights of a given layer.

We provide a modified version of a simple deep convolutional neural network and a resnet (25) with its convolutional layers' weights trained on the hypersphere. They were trained respectively on the MNIST (41) and (38) datasets.

### 3.3 HYPERBOLIC SPACE - USE CASE REVIEWS IN MACHINE LEARNING

We now focus on the applications of hyperbolic spaces in the machine learning literature. Hyperbolic spaces arise in information and learning theory. Indeed, the space of univariate Gaussians endowed with the Fisher metric densities is a hyperbolic space (13). This characterization is used in various fields, such as in image processing, where each image pixel is represented by a Gaussian distribution (3), or in radar signal processing where the corresponding echo is represented by a stationary Gaussian process (4).

The hyperbolic spaces can also be seen as continuous versions of trees and are therefore interesting when learning hierarchical representations of data (50). Hyperbolic geometric graphs (HGG) have also been suggested as a promising model for social networks, where the hyperbolicity appears through a competition between similarity and popularity of an individual (52).

### 3.4 GEOMSTATS USE CASE - VISUALIZATION ON THE HYPERBOLIC SPACE $H_2$

We present the visualization toolbox provided in `geomstats`. This toolbox plays an educational role by enabling users to test their intuition on Riemannian manifolds. Users can run and adapt the examples provided in the `geomstats/examples` folder of the supplementary materials. For example, we can visualize the hyperbolic space $H_2$ through the Poincare disk representation, where the border of the disk is at infinity. The user can then observe how a geodesic grid and a geodesic square are deformed in the hyperbolic geometry on Figure 2.

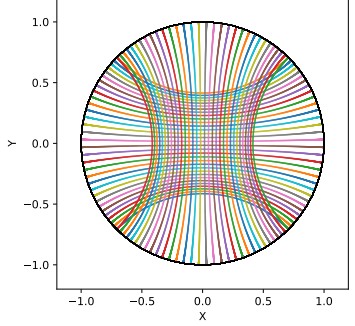 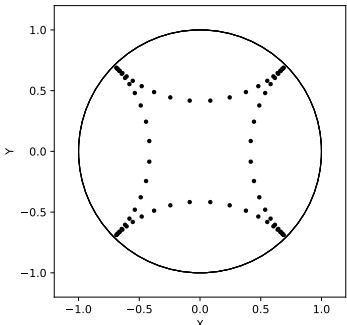

Figure 2: Left: Regular geodesic grid on the Hyperbolic space $H^2$ in Poincare disk representation. Right: Geodesic square on the Hyperbolic space $H_2$, with points regularly spaced on the geodesics defining the square's edges.

## 4 MANIFOLD OF SYMMETRIC POSITIVE DEFINITE (SPD) MATRICES

The previous section dealt with the Hypersphere and the Hyperbolic space, that is, manifolds embedded in flat spaces. We now increase the geometric complexity and consider a manifold embedded in the general linear group of invertible matrices. The manifold of symmetric positive definite (SPD) matrices in $n$ dimensions is defined as:

$$SPD = \left\{ S \in \mathbb{R}_{n \times n} : S^T = S, \forall z \in \mathbb{R}^n, z \neq 0, z^T S z > 0 \right\}. \tag{3}$$

The class `SPDMatricesSpace` inherits from the class `EmbeddedManifold` and has an `embedding_manifold` attribute which stores an object of the class `GeneralLinearGroup`.

We equip the manifold of SPD matrices with an object of the class `SPDMetric` that implements the affine-invariant Riemannian metric of (54) and inherits from the class `RiemannianMetric`.

## 4.1 SPD Matrices Manifold - Use Cases in Machine Learning

SPD matrices are ubiquitous in machine learning across many fields (12), either as input or output to the problem. In diffusion tensor imaging (DTI) for instance, voxels are represented by "diffusion tensors" which are 3x3 SPD matrices. These ellipsoids spatially characterize the diffusion of water molecules in the tissues. Each DTI thus consists in a field of SPD matrices, which are inputs to regression models. In (66) for example, the authors use an intrinsic local polynomial regression applied to comparison of fiber tracts between HIV subjects and a control group. Similarly, in functional magnetic resonance imaging (fMRI), it is possible to extract connectivity graphs from a set of patients' resting-state images' time series (60; 64; 30)– a framework known as brain connectomics. The regularized graph Laplacians of the graphs form a dataset of SPD matrices. They represent a compact summary of the brain's connectivity patterns which is used to assess neurological responses to a variety of stimuli (drug, pathology, patient's activity, etc.). SPD matrices can also encode anatomical shape changes observed in images. The SPD matrix $J^T J^{1/2}$ represents the directional information of shape change captured by the Jacobian matrix $J$ at a given voxel (23).

More generally speaking, covariance matrices are also SPD matrices which appear in many settings. We find covariance clustering used for sound compression in acoustic models of automatic speech recognition (ASR) systems (58) or for material classification (19) among others. Covariance descriptors are also popular image or video descriptors (24).

Lastly, SPD matrices have found applications in deep learning, where they are used as features extracted by a neural network. The authors of (22) show that an aggregation of learned deep convolutional features into a SPD matrix creates a robust representation of images that enables to outperform state-of-the-art methods on visual classification.

## 4.2 Geomstats Use Case - Connectivity Graph Classification

We show through a concrete brain connectome application how `geomstats` can be easily leveraged for efficient supervised learning on the space of SPD matrices. The folder `brain_connectome` of the supplementary materials contains the implementation of this use case.

We consider the fMRI data from the 2014 MLSP Schizophrenia Classification challenge[2], consisting of the resting-state fMRIs of 86 patients split into two balanced categories: control vs people suffering schizophrenia. Consistently with the connectome literature, we tackle the classification task by using a SVM classifier on the precomputed pairwise-similarities between subjects. The critical step lies in our ability to correctly identify similar brain structures, here represented by regularized Laplacian SPD matrices $\hat{L} = (D - A) + \gamma I$, where A and D are respectively the adjacency and the degree matrices of a given connectome. The parameter $\gamma$ is a regularization shown to have little effect on the classification performance (15).

Following two popular approaches in the literature (15), we define similarities between connectomes through kernels relying on the Riemannian distance $d_R(\hat{L}_1, \hat{L}_2) = || \log(\hat{L}_1^{-1/2}.\hat{L}_2.\hat{L}_1^{-1/2})||_F$ and on the log-Euclidean distance, a computationally-lighter proxy for the first: $d_{LED}(\hat{L}_1, \hat{L}_2) = || \log_I(\hat{L}_2) - \log_I(\hat{L}_1)||_F$. In these formulae, $\log$ is the matrix logarithm and $F$ refers to the Frobenius norm. Both of these similarities are easily computed with `geomstats`, for example the Riemannian distance is obtained through `metric.squared_dist` where `metric` is an instance of the class `SPDMetric`.

Figure 3 (left) shows the performance of these similarities for graph classification, which we benchmark against a standard Frobenius distance. With an out-of-sample accuracy of 61.2%, the log-Euclidean distance here achieves the best performance. Interestingly, the affine-invariant Riemannian distance on SPD matrices is the distance that picks up the most differences between connectomes. While both the Frobenius and the log-Euclidean recover only very slight differences between connectomes –placing them almost uniformly afar from each other–, the Riemannian distance exhibits greater variability, as shown by the clustermap in Figure 3 (right). Given the ease of implementation

---

[2]Data openly available at `https://www.kaggle.com/c/mlsp-2014-mri`

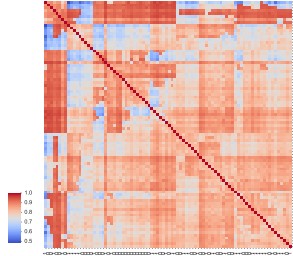

| Distance | Accuracy | F1-Score |
|---|---|---|
| Riemannian | 30.8% | **47.1** |
| Log Euclidean | **62.5** | 36.4 |
| Frobenius | 46. 2% | 0.00 |

Figure 3: Left: Connectome classification results. Right: Clustermap of the recovered similarities using the Riemannian distance on the SPD Manifold. We note in particular the identification of several clusters (red blocks on the diagonal)

of these similarities with `geomstats`, comparing them further opens research directions for in-depth connectome analysis.

# 5 LIE GROUPS $SO(n)$ AND $SE(n)$ - ROTATIONS AND RIGID TRANSFORMATIONS

The previous sections tackled the case of manifolds which were embedded in other manifolds, the latter being either flat or endowed with a Lie group structure. We now turn to manifolds that are Lie groups themselves. The special orthogonal group $SO(n)$ is the group of rotations in $n$ dimensions defined as

$$SO(n) = \left\{ R \in \mathbb{R}_{n \times n} : R^T.R = Id_n \text{ and } \det R = 1 \right\}. \tag{4}$$

The special Euclidean group $SE(n)$ is the group of rotations and translations in $n$ dimensions defined by its homegeneous representation as

$$SE(n) = \left\{ X \in \mathbb{R}_{n \times n} \quad | \quad X = \left[ \begin{array}{c|c} R & t \\ \hline 0 & 1 \end{array} \right], t \in \mathbb{R}^n, R \in SO(n) \right\} \tag{5}$$

The classes `SpecialOrthogonalGroup` and `SpecialEuclideanGroup` both inherit from the classes `LieGroup` and `EmbeddedManifold`, as embedded in the General Linear group. They both have an attribute `metrics` which can store a list of metric objects, instantiations of the class `InvariantMetric`. A left- or right- invariant metric object is instantiated through an inner-product matrix at the tangent space at the identity of the group.

## 5.1 LIE GROUPS $SO(n)$ AND $SE(n)$ - USE CASES IN MACHINE LEARNING

Lie groups $SO(n)$ and $SE(n)$ for data and parameters representation are also a popular object in many a machine learning problem. In 3D, $SO(3)$ and $SE(3)$ appear naturally when dealing with articulated objects. A spherical robot arm is an example of an articulated object whose positions can be modeled as the elements of $SO(3)$. The human spine can also be modeled as an articulated object where each vertebra is represented as an orthonormal frame that encodes the rigid body transformation from the previous vertebra (5; 7). In computer vision, elements of $SO(3)$ or $SE(3)$ are used to represent the orientation or pose of cameras (33). Supervised learning algorithm predicting such orientations or poses have numerous applications for robots and autonomous vehicles which need to localize themselves in their environment.

Lastly, the Lie group $SO(n)$ and its extension to the Stiefel manifold are found very useful in the training of deep neural networks. The authors of (29) suggest to constrain the network's weights on a Stiefel manifold, i.e. forcing the weights to be orthogonal to each other. Enforcing the geometry significantly improves performances, reducing for example the test error of wide residual network on CIFAR-100 from 20.04% to 18.61% .

## 5.2 GEOMSTATS USE CASE - GEODESICS ON $SO(3)$

Riemannian geometry can be easily integrated for machine learning applications in robotics using `geomstats`. We demonstrate this by presenting the interpolation of a robot arm trajectory by geodesics. The folder `robotics` of the supplementary materials contains the implementation of this use case.

In robotics, it is common to control a manipulator in Cartesian space rather than configuration space. This allows for a much more intuitive task specification, and makes the computations easier by solving several low dimension problems instead of a high dimension one. Most robotic tasks require to generate and follow a position trajectory as well as an orientation trajectory.

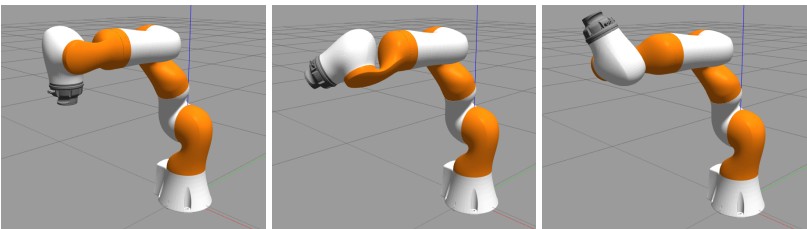

Figure 4: A Riemannian geodesic computed with the canonical bi-invariant metric of $SO(3)$, applied to the extremity of the robotic arm.

While it is quite easy to generate a trajectory for position using interpolation between several via points, it is less trivial to generate one for orientations that are commonly represented as rotation matrices or quaternions. Here, we show that we can actually easily generate an orientation trajectory as a geodesic between two elements of $SO(3)$ (or as a sequence of geodesics between several via points in $SO(3)$). We generate a geodesic on $SO(3)$ between the initial orientation of the robot and its desired final orientation, and use the generated trajectory as an input to the robot controller. The trajectory obtained is illustrated in Figure 4.

This opens the door for research at the intersection of Riemannian geometry, robotics and machine learning. We could ask the robot arm to perform a trajectory towards an element of $SE(3)$ or $SO(3)$ predicted by a supervised learning algorithm trained for a specific task. The next subsection presents the concrete use case of training a neural network to predict on Lie groups using `geomstats`.

### 5.3 GEOMSTATS USE CASE - DEEP LEARNING PREDICTIONS ON $SE(3)$

We show how to use `geomstats` to train supervised learning algorithms to predict on manifolds, specifically here: to predict on the Lie group $SE(3)$. This use case is presented in more details in the paper (28) and the open-source implementation is given. The authors of (28) consider the problem of pose estimation that consists in predicting the position and orientation of the camera that has taken a picture given as inputs.

The outputs of the algorithm belong to the Lie group $SE(3)$. The `geomstats` package is used to train the CNN to predict on $SE(3)$ equipped with a left-invariant Riemannian metric. At each training step, the authors of (28) use the loss given by the squared Riemannian geodesic distance between the predicted pose and the ground truth. The Riemannian gradients required for back-propagation are given by the closed forms implemented in `geomstats`.

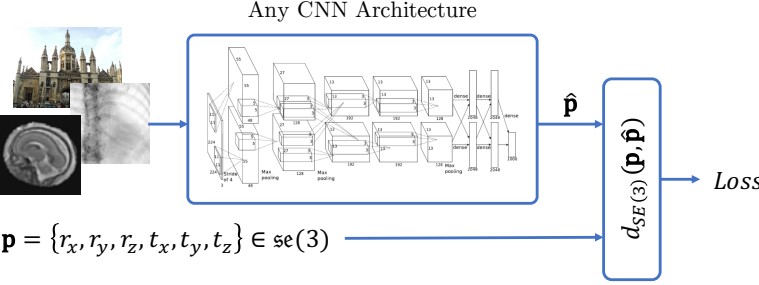

Figure 5: Image courtesy of (28). CNN with a squared Riemannian distance as the loss on $SE(3)$.

The effectiveness of the Riemannian loss is demonstrated by experiments showing significative improvements in accuracy for image-based 2D to 3D registration. The loss functions and gradients

provided in `geomstats` extend this research directions to CNN predicting on other Lie groups and manifolds.

## 6 CONCLUSION AND OUTLOOK

We introduce the open-source package `geomstats` to democratize the use of Riemannian geometry in machine learning for a wide range of applications. Regarding the geometry, we have presented manifolds of increasing complexity: manifolds embedded in flat Riemannian spaces, then the case of the SPD matrices space and lastly Lie groups with invariant Riemannian metrics. This provides an educational tool for users who want to delve into Riemannian geometry through a hands-on approach, with intuitive visualizations for example in subsections 3.4 and 5.2.

In regard to machine learning, we have presented concrete use cases where inputs, outputs and parameters belong to manifolds, in the respective examples of subsection 4.2, subsection 5.3 and subsection 3.2. They demonstrate the usability of `geomstats` package for efficient and user-friendly Riemannian geometry. Regarding the machine learning applications, we have reviewed the occurrences of each manifold in the literature across many different fields. We kept the range of applications very wide to show the many new research avenues that open at the cross-roads of Riemannian geometry and machine learning.

`geomstats` implements manifolds where closed-forms for the Exponential and the Logarithm maps of the Riemannian metrics exist. Future work will involve implementing manifolds where these closed forms do not necessarily exist. We will also provide the `pytorch` backend.

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
