# OpenReview forum: "Geomstats: a Python Package for Riemannian Geometry in Machine Learning"
_ICLR.cc/2019/Conference_

### Official Review · AnonReviewer3 · 2018-11-02
**geomstats has a potential to generate large impact to the community**

**Rating:** 8
**Confidence:** 2

**Review:**

This white paper presents the geomstats package. The package provides tools for Riemannian modelization and
optimization over manifolds. Especially, the package supports several important manifolds: hyperspheres, hyperbolic spaces, spaces of SPD matrices or Lie groups of transformations.

Pros:
1. the paper shows ever use cases of machine learning with manifolds. These use cases are concrete and representative.
2. the code in the package is extensively tested.

Cons:

There is no discussion about the scalability of the package.

---

### Official Review · AnonReviewer1 · 2018-11-03
**More of a software document than a scientific paper!**

**Rating:** 3
**Confidence:** 5

**Review:**

This paper introduces Geomstats, a geometric toolbox for machine learning on Riemannian manifolds. In comparison to previous packages such as manopt and pymanopt, the paper claims that the proposed software provide more efficient implementations, and is integrated with deep learning backends. Several potential applications settings for the software are explored, introducing some performance gains on specific problems when one resorts to the geometry of the space. An example setting for deep learning on SE(3) is also presented.

Strengths: The use of such a toolbox could be a significant step to leveraging geometric models in deep learning.

Weakness:

1. The paper is written as more of a software document than a scientific paper. Several well-known manifolds are presented in the setting of Geomstats, however what is lacking is a treatment of some of the internals of how backpropagation can be implemented effectively for such a toolbox. For example, many of the Riemannian geometric algorithms need to resort to numerical algorithms such as SVD and such; how effective or feasible is automatic differentiation in dealing with such cases?

2. The paper discusses computational advantages in the introduction -- however such advantages are not quantitatively analyzed across different platforms or prior softwares.

Overall, it is not clear why this work needs to be treated as a scientific paper? It appears to be more of a tutorial on the use of the proposed software.

---

### Official Review · AnonReviewer2 · 2018-11-05
**Novelty not clear**

**Rating:** 4
**Confidence:** 5

**Review:**

Summary:
The paper is well written and easy to follow. The paper proposes a Python package for optimization and applications on Riemannian manifolds.

Comments:

C1.
The main concern of the package is on novelty. That there exist other packages, e.g., Pymanopt [1], Manopt [2], ROPTLIB [3], which do a similar job as Geomstats. It is hard to understand from the paper on what is the key contribution of the present package. The paper does try to highlight the differences between the Goemstats package and others by emphasizing the lack of “choice of other metrics and high computational costs of” others. This is, however, not shown in the paper as to how Geomstats is better in terms of computational complexity. On the choice of the metrics too, it should be noted that all the new geometries of the manifolds (for different metrics) can be easily added to the toolboxes [1,2,3]. Those toolboxes are modular and come with a lot of solvers and have already been used in many large-scale applications.

Having said that, one key strength of the Geomstats package is that it can be used in a deep learning framework in a relatively straightforward manner. This is mentioned in the paper but is not properly emphasized. However, here also, the paper does not say what algorithms can be used as part of Geomstats (except the stochastic gradient descent optimizer).

Overall, I got the impression that the proposed package positions itself as a package of Riemannian manifolds (and all of the differential geometric notions). If that is so, then this is problematic. Pymanopt and others already do that fairly well, and consequently, the justification for a new package on that basis is difficult.

[1] https://pymanopt.github.io/
[2] https://www.manopt.org/
[3] https://www.math.fsu.edu/~whuang2/Indices/index_ROPTLIB.html


C2.
The citations are not properly rendered. It is hard to distinguish between the equations in the paper and the references in the paper.

C3.
Instead of multiple use cases in the paper, could the paper focus one particular use case to show the important functionalities of Geomstats? Currently, it seems that the paper is more about surveying where all Riemannian geometry is useful.

---

### Official Review · AnonReviewer4 · 2018-11-08
**Nice package but with limited novelty and largely undemonstrated advantages to existing frameworks**

**Rating:** 4
**Confidence:** 4

**Review:**

The paper introduces the software package geomstats which provides simple use of Riemannian manifolds and metrics within machine learning models. Like theanogeometry, geomstats provides a backend for fast computation. Instead of theano, they interface tensorflow and numpy.

The core problem the author’s have to argue against is the existence of various other packages like pymanopt (which are mentioned in the paper) providing similar functionality.

The main advantage to pymanopt is stated to be lower computational cost. Unfortunately, this is not evaluated empirically. Pymanopt similarly provides the option to provide the cost function with tensorflow and uses numpy/scipy internally, therefore also making use of vectorization. A favorable empirical comparison would have been a compelling case for geomstats. While geomstats provides some more metrics than pymanopt, it lacks in other areas in comparison. Such metrics could be added relatively easily to pymanopt (or some of the other competing libraries).

Truly novel is the update for Keras which allows Riemannian gradient descent on parameters living on manifolds. Unfortunately, this is not directly shown and discussed further in the paper, but the reader is referred to the code base. While plenty of examples were provided in the supplementary material, I’d have preferred to see specific example(s) being shown and discussed in the paper. In the end, the main paper alone only gives an overview of what exists, but gives me no idea on how the package is used.

In parts, the paper reads more like an argument in favor of Riemannian modelization and optimization, instead of advocating for the specific package. While it is very important to demonstrate potential applications for the framework - an area in which this paper excelled - other, more important parts (mentioned above), were omitted because of it. Similarly, a lot of time is needlessly spent on defining well-known manifolds.

On the formal side, the formatting of the citations within the text don’t adhere to the official style guide which prescribes the use of authors’ last names and year.

Overall, the software package seems to provide nice functionality with integration into a currently popular machine learning framework, but it’s novelty compared to existing software packages is limited. The novel parts (performance improvements, integration with keras) are not sufficiently demonstrated in the paper.

---

### Public Comment · (anonymous) · 2018-11-01
**How is tensorflow and keras modified?**

This is not an official review, just a few comments from an interested reader.

The contribution in the present paper is a piece of software, i.e. no new methodology is presented. This seem to be in line with the ICLR CfP, though personally I would prefer fairly few "software only" papers at conferences I attend.

I wanted to ask the authors, how tensorflow and keras are modified? The paper states "We provide modified versions of keras and tensorflow..." Does this mean that you have forked these libraries and modified them? Does that mean that using "geomstats" imply not using my own tensorflow/keras installation?

---

### Author Response · Authors · 2018-11-27
**Geomstats: beyond optimization, a flexible package for modelization and machine learning on Riemannian manifolds**

We thank the reviewers for their constructive feedback which has shown us the need to clearly highlight the impact and scope of our contribution. We answer their concerns regarding the novelty, practicality and use of the package. In summary, we feel that the reviewers have focused too much on the “Riemannian optimization” side of Geomstats, thus partly neglecting our package’s novelty which is its completeness and flexibility in terms of Riemannian geometry.

# Novelty with respect to other softwares

R1 and R2 expressed concerns on the novelty of Geomstats in light of other existing softwares to perform optimization on manifolds. We contend that Geomstats was built with a completely different purpose than these softwares: its objective is to foster research and use of “Riemannian” data models. This translates into a set of unique specificities:

1. Geometry : the package’s object-oriented design of Riemannian geometry is in itself a (novel) mathematical contribution. Having common categories of objects and operations for differential geometric structures such as affine connection spaces, Riemannian spaces, Lie groups, etc. is not easy to realize and requires an important effort to verify if theorems and notions continue to hold for each structure (done in our unit tests). This is not at all present in any other packages.

Our package is also the first to provide a flexible choice of Riemannian metrics and corresponding methods: metrics and left- or right- invariant metrics on Lie groups are not implemented in any of the other packages. R1 argues that these metrics could be easily added to Pymanopt and we disagree. The metrics would come within Pymanopt only for the purpose of optimization, which considerably restricts their use. Moreover, they wouldn’t come with exponential and logarithm maps, inner products at each tangent space, tangent pca, etc., because Pymanopt focuses on efficient optimization rather than completeness.

Our package thus positions itself as a package of Riemannian modelization, with significantly more geometric features than Pymanopt which reflect this specific purpose.

2. Flexibility: Geomstats is uniquely flexible and modular enough to allow researchers to contribute their code. For instance, a researcher recently contributed an algorithm for optimal quantization on Riemannian manifolds, making use of our exponential maps and enriching the package with new methods. This could not have been implemented in the context of Pymanopt. This flexibility is also reflected in the multiple use of Riemannian geometry that Geomstats permits. It is in particular the only package that draws a heavy accent on statistical analysis of manifolds (new notion of mean, variance, etc.), which come readily implemented in Geomstats--making it a unique gateway to principled inference on manifolds.

It seems that the reviewers would have preferred one specific objective for Geomstats (e.g. optimization, or one specific example of its use), but this would have been against its purpose which is meant to be broad, as a fertile ground for geometry in machine learning.

3. Practicality

Geomstats is in python and fully integrated with numpy, while Manopt and  Roptlib are being mainly devised for Matlab or Julia, thus severely  hindering the adoption of Riemannian geometry by the machine learning community.

Geomstats is yet the only Riemannian software directly integrated with Tensorflow and Keras, allowing GPU computations and thus computational efficiency and the use of Riemannian geometry in large scale problems, as shown in Sec 5.3 and [1]. Using Geomstats in this large scale deep learning application improves the performance in all image metrics ([1] Table 2), while training time is comparable (communication from [1]).

# Software document

R3 considers our submission as more of a software document than a scientific paper. We introduce a software meant for machine learning research, being conform to ICLR’s call for papers on “software platforms”.

We thank the reviewers again for the attention given to our contribution and hope that we have lifted their objections to our manuscript. We believe this package is of general interest with a potentially broad impact for the ML community and should get visibility at ICLR conference to foster collaborative work around geometry.

[1] Hou, B., Miolane, N., Khanal, B., Lee, M., Alansary, A., McDonagh, S., Hajnal, J., Ruecket, D., Glocker, B., Kainz, B.: Deep pose estimation for image-based registration. MICCAI 2018.

---

### Meta-Review · Area_Chair1 · 2018-12-12
**perhaps for another venue?**

**Confidence:** 5
**Recommendation:** Reject

**Metareview:**

Learning on Riemannian manifolds can be easily done  with this Python package.  Considering the recent work on these in latent-variable models, the package can be quite a useful approach.

But its novelty is disputed.  In particular Pymanopt is a package that does mostly the same, even though that may be computationally more expensive.  The merits of Geomstats vs. Pymanopt is not clarified.  But be that as it may, there is interest amongst the reviewers for the software package.

In the end, too, it's not uniformly agreed upon that a software-describing paper fits ICLR.